# A New Density-Based Clustering Method Considering Spatial Distribution of Lidar Point Cloud for Object Detection of Autonomous Driving

**Caihong Li [1], Feng Gao [1,2,*], Xiangyu Han [1] and Bowen Zhang [1]**

[1] College of Mechanical and Vehicle Engineering, Chongqing University, Chongqing 400044, China; LiCaihong@cqu.edu.cn (C.L.); 20186102@cqu.edu.cn (X.H.); 20171113034t@cqu.edu.cn (B.Z.)

[2] Sichuan Research Institute, Shanghai Jiao Tong University, Chengdu 610200, China

* Correspondence: gaofeng1@cqu.edu.cn; Tel.: +86-189-9618-8196

**Abstract:** Lidar is a key sensor of autonomous driving systems, but the spatial distribution of its point cloud is uneven because of its scanning mechanism, which greatly degrades the clustering performance of the traditional density-based spatial clustering of application with noise (DSC). Considering the outline feature of detected objects for intelligent vehicles, a DSC-based adaptive clustering method (DAC) is proposed with the adoption of an elliptic neighborhood, which is designed according to the distribution properties of the point cloud. The parameters of the ellipse are adaptively adjusted with the location of the sample point to deal with the uniformity of points in different ranges. Furthermore, the dependence among different parameters of DAC is analyzed, and the parameters are numerically optimized with the KITTI dataset by considering comprehensive performance. To verify the effectiveness, a comparative experiment was conducted with a vehicle equipped with three IBEO LUX8 lidars on campus, and the results show that compared with DSC using a circular neighborhood, DAC has a better clustering performance and can notably reduce the rate of over-segmentation and under-segmentation.

**Keywords:** autonomous driving; object detection; lidar detection; clustering method

## 1. Introduction

It is obvious that automated vehicles have a profound impact on human social activities and lifestyles [1,2], in which object detection plays an important role to ensure driving safety, especially in the location estimation and the trajectory prediction of objects [3]. Lidar has been gradually applied in production vehicles because of its advantages of high precision, long range and shape detection [4,5]. One example of such an application is the lidar-assisted automatic obstacle avoidance system, which recognizes the location and size of obstacles to help decide whether to try an overtaking maneuver [6], and this system still plays an auxiliary role on foggy and rainy days. To realize object detection with the lidar point cloud, it is important to correctly cluster all points belonging to the same object together [7], which is further used to estimate contours, position, object type and other information [8].

The point cloud density is dense in close distance, while it is sparse in far distance. To improve the clustering performance of lidar caused by its unbalanced distribution data, Hasecke et al. clustered the point cloud by semantic segmentation using the neighborhood relationship of the 3D measurement information [9]. Furthermore, a camera was combined with lidar to realize the complementation of the spatial and color information [10–12]. However, these advanced methods require huge amounts of computing resources, and an adequate dataset is necessary to train the algorithms. Comparatively, the improved traditional clustering method is still an effective and practical way for considering the computation ability of the onboard unit and the collection difficulty of a complete dataset.

According to different clustering principles, the traditional clustering methods are mainly divided into the partition-based method [13–15], the hierarchical method [16,17] and the density-based method [18–20]. The fundamental of the partition-based method is that the lidar points are assigned to clusters by minimizing the sum of the distance between the points in a cluster, meanwhile maximizing it among different clusters [14], e.g., k-medoids [15]. The number of clusters should be defined in advance when using this method. The hierarchical clustering method does not have this limitation, but it is unsuitable for large point clouds due to its high computational complexity [17]. Compared with the aforementioned two methods, density-based clustering can detect objects with arbitrary shapes by consuming acceptable computation resources. Accordingly, a density-based spatial clustering of application with noise (DSC) has been applied to the lidar detection of objects for autonomous driving systems [20].

Due to the inhomogeneous distribution of the point cloud, DSC with fixed parameters has a comparatively low detection performance [21], because the point cloud density has a significant influence on the parameter values [22,23]. There are mainly two ways to deal with this:

(a)  Hierarchical approach

The clustering process is divided into two stages. In the first stage, several sub-clusters and their parameters are determined by Euclidean distance [24] or k-means [25]. Then, the DSC strategy is used to categorize these sub-clusters into noise or different objects.

(b)  Space partition

The detected area is evenly divided into several regions beforehand. Then, the optimal parameters of each region are calculated by the geometric structure of boundary points [26], or the original partition is further optimized by considering the change rate of point density in different regions [27]. Finally, DSCs with different parameters are adopted for each region to realize object classification.

Generally, the former has higher computational complexity, which is a challenge for the real-time application of the onboard units. Conversely, the latter is computationally efficient, but the clustering performance greatly depends on the initial partition of the detected area, continuity and smoothness between sub-regions [28].

For lidar object detection in autonomous driving systems, a DSC-based adaptive clustering (DAC) method is proposed in this paper by considering the spatial distribution of the point cloud to improve the overall clustering results of traditional DSC. According to the analysis results of the spatial distribution characteristics of the point cloud based on the point distance model of the object contour, an ellipse neighborhood, whose parameters are adaptively adjusted by the sample point, is designed. Furthermore, the relationship among different parameters of DAC is analyzed and designed numerically using the KITTI dataset, which has provided an open-access dataset and standard evaluation mean for road area detection. The effectiveness of DAC is validated by comparative tests, which were conducted on campus using a vehicle equipped with three IBEO LUX8 lidars.

The rest of this paper is organized as follows: Section 2 analyzes the problems of traditional DSC. In Section 3, the DAC method is proposed based on the spatial distribution of point cloud. Parameters of DAC are analyzed in Section 4. Section 5 validates the effectiveness of DAC, and Section 6 concludes the paper.

## 2. Problem Analysis

The process of DSC consists of two stages as shown in Figure 1 [29]. One is the pre-processing stage, in which the original point cloud is reduced by removing useless data to achieve a better performance. The other is the clustering stage, where points belonging to the same object are clustered together to achieve object detection.

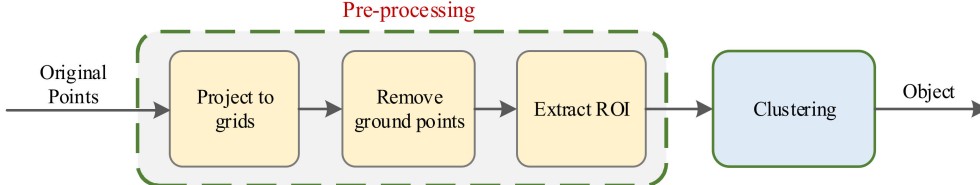

**Figure 1.** Diagrams of DSC-based clustering method.

In this study, the original points are first projected onto a grid map [30,31], and then the ground points are removed by using the maximum height difference [32]. Finally, Hough transform is used to detect roadsides to obtain the dynamic region of interest (ROI) [33]. The reduction effect of the lidar points by the pre-processing is shown in Figure 2 as an example.

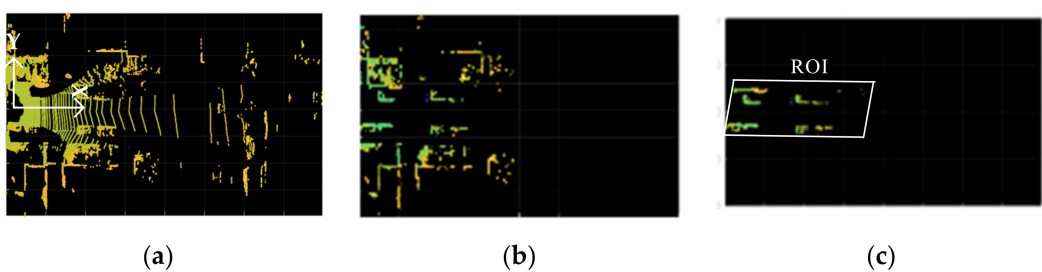

(a)                                        (b)                                        (c)

**Figure 2.** Effect of pre-processing. (**a**) Projection of original points; (**b**) removal of ground points; and (**c**) extraction of dynamic ROI.

In the clustering stage, because the spatial density of the point cloud varies with the point position, over-segmentation and under-segmentation easily occur if a fixed value is selected as the clustering parameter. As shown in Figure 3, when two objects are close to the lidar (Figure 3a), a small clustering radius is required to distinguish them. On the contrary, when they are far away (Figure 3b), the points belonging to the same object may be separated into different classes if the small clustering radius is still applied, because the point distribution is sparse in the far region. In this condition, a large clustering radius is preferred to avoid over-segmentation.

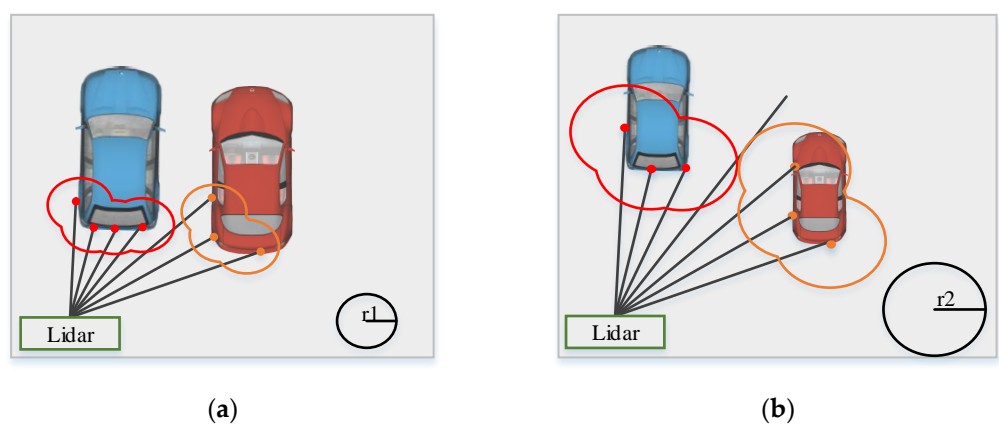

(a)                                        (b)

**Figure 3.** Problem analysis of clustering in different regions. (**a**) Near region and (**b**) far region.

To show this problem more intuitively, the traditional DSC was tested on the KITTI dataset [34], with the clustering radius and the minimum number of points (denoted by MinPts) set to 0.15 m and 5, respectively. Some typical wrong segmentations are shown in Figure 4.

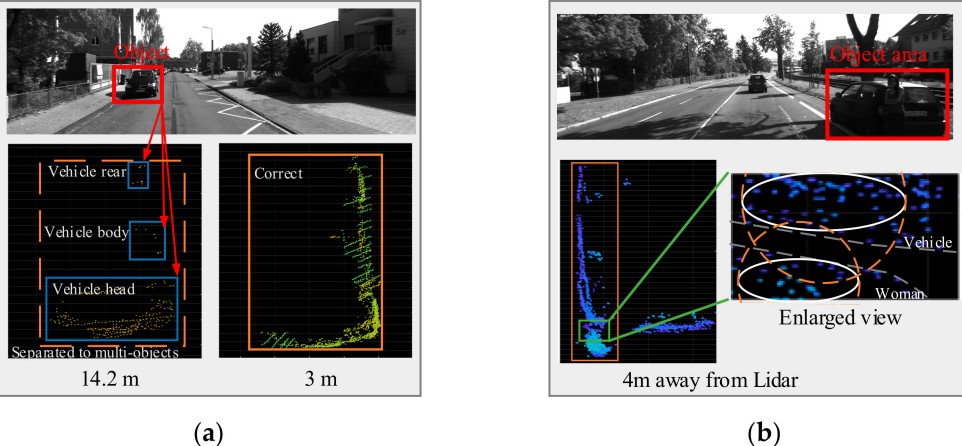

**Figure 4.** Typical wrong segmentations. (**a**) Over-segmentation and (**b**) under-segmentation.

In Figure 4a, the vehicle is detected correctly when it is 3 m away from the lidar, but it is wrongly classified into three objects when it lies at 14.2 m due to the distance between the points becoming sparse. In Figure 4b, a woman stands too close to the vehicle, and they are wrongly classified as the same object. As can be seen from the enlarged view, their points are very close, and a smaller searching radius is required to distinguish them. Considering that the point spacing between objects is different from that within objects, an anisotropic shape may also distinguish them correctly. To compare the clustering effects of isotropic and anisotropic shapes, circles and ellipses, whose diameter is the same as the long axis, are applied to the two objects in the enlarged image, and the result shows that the circular neighborhood is more prone to under-segmentation than the ellipse.

From the above analysis, the following is found:

(a) The traditional DSC with fixed parameter values cannot make a good balance between the clustering performances of far and near objects and wrong segmentations easily happen. A different clustering radius is necessary for objects in different regions.

(b) For the same object, the point distribution of its contour along the longitudinal and lateral directions is not the same. This causes the circular neighborhood to cover much of the blank area, which easily leads to under-segmentation.

To solve such problems for a better clustering performance, a new searching neighborhood is designed according to the spatial distribution analysis of the lidar points, and an online algorithm is proposed to adjust the parameter of the neighborhood for the adaption to objects in different regions.

## 3. DAC Method

A new clustering method is designed in this section based on the spatial distribution analysis of the point cloud using the point distance model of the object contour.

### 3.1. Spatial Distribution Analysis of Point Cloud

The typical outlines of detected objects are "—", "|" and "L" in autonomous driving systems [35]; thus, only the point distances along the lateral and longitudinal directions are considered. The point distance model is established according to the scanning mechanism of lidar depicted by Figure 5, where only half of the region is shown because of the symmetry of the scanning area.

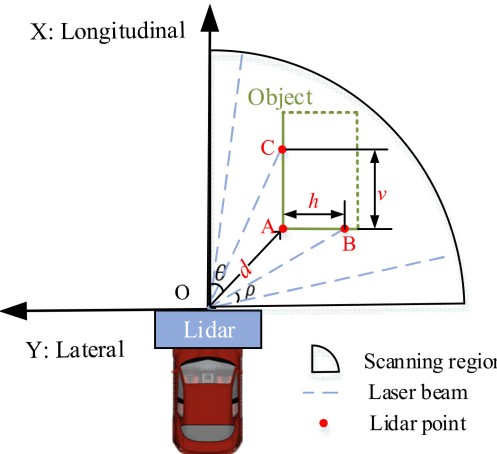

**Figure 5.** Point distance model.

The point spacing along the lateral and longitudinal directions is calculated by

$$h = \frac{\sin\rho}{\cos(\theta+\rho)}\cdot d, v = \frac{\sin\rho}{\sin(\theta-\rho)}\cdot d \tag{1}$$

where $h$ and $v$ are the distance between two adjacent points along the lateral and longitudinal directions, respectively; $\rho$ is the horizontal angular resolution of the lidar; $\theta$ is the azimuth angle of the point; and $d$ is the distance from the point to the lidar.

It can be seen from (1) that both $h$ and $v$ increase linearly with $d$, and their coefficient relationship is explored by calculating the ratio as Equation (2), in which $\rho$ is too small to be approximately ignored (For IBEO, $\rho = 0.125°$).

$$\frac{h}{v} = \frac{\sin(\theta-\rho)}{\cos(\theta+\rho)} \approx \tan\theta \tag{2}$$

For points on diagonal lines, $h = v$, it is appropriate to apply a circle as the neighborhood shape. However, in other regions, $h$ and $v$ are obviously different; thus, a shape with different radii in two directions is needed to avoid the wrong segmentation caused by the isotropic neighborhood. Thus, an ellipse is selected as the shape of the searching neighborhood, which had its superiority verified in Figure 4b.

As the point spacing increases with the distance from the lidar, the ellipse neighborhood should be dynamically adjusted with the position of the sample point, ensuring all points can be grouped into the correct classes. To obtain the suitable ellipse area, the ranges of $h$ and $v$ are calculated by (1) first, which provides the basis for constructing the connection between point spacing and ellipse parameters. The range of the point distance is depicted in Figure 6, where the lateral range is less than 8 m, because, for autonomous driving, the detected objects are located in two adjacent lanes with a width of 3.5–3.8 m [36].

The following can be observed from Figure 6:

(a) In this detected region, most of $h$ is less than 0.273 m, while there are abrupt changes at the poles of $\theta$, $h_{max1} = 0.262$ m when $\theta \in (0°,\ 45°)$ and $h_{max2} = 0.597$ m for $\theta \in (45°,\ 90°)$, but their orders of magnitude are still the same. That is, $h$ has a very stable variation; thus, the length of the axis associated with it can be considered as a constant.

(b) On the contrary, the variation of $v$ is more pronounced than that of $h$, and the correlation between $v$ and $d$ is stronger. The maximum amplitude of $v$ can reach 13.953 m, of which 95% is less than 5.81 m, but it is still too large to be ignored compared with the size of the detected object. To ensure that an appropriate neighborhood area is obtained for each sampling point, the axis related to the longitudinal point spacing must be variable.

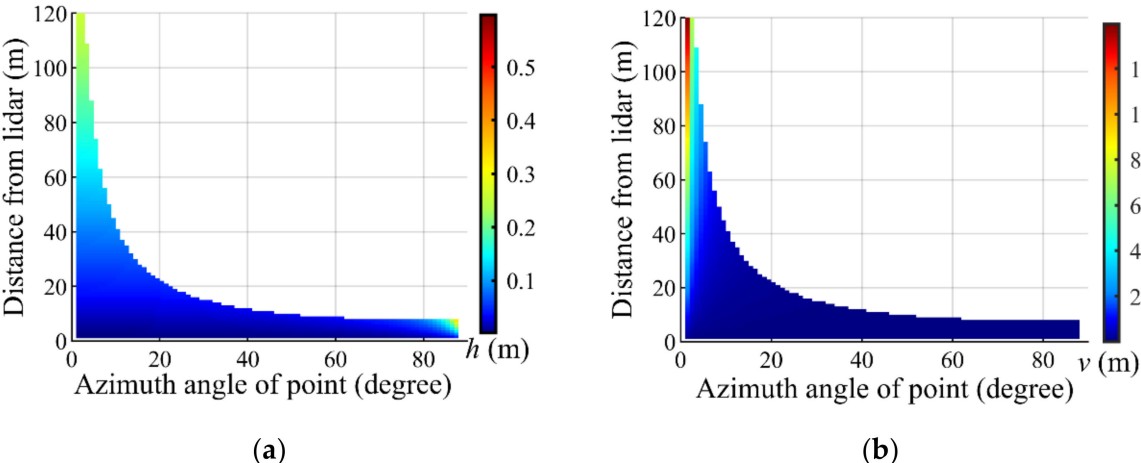

**Figure 6.** Ranges of distances between two adjacent points. (**a**) Lateral point spacing and (**b**) longitudinal point spacing.

### 3.2. DAC Design

An ellipse with fixed and variable axes, as described in Section 3.1, is applied to the proposed clustering method. If the scanning points of the object are arranged at the edges of the two sides of the object, as shown in Figure 7, two ellipses with a different major axis are needed to gather these points correctly, because the ratio of these two point spacings cannot determine the direction of the major axis. For the convenience of description, the semi-major and semi-minor of the ellipse are defined by longitudinal spacing $v$ and lateral spacing $h$, respectively.

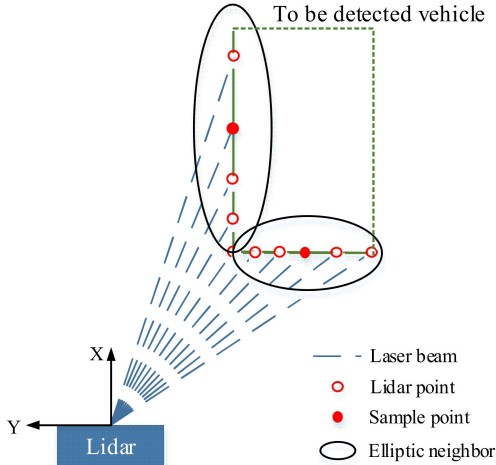

**Figure 7.** Clustering of points in different directions, where *MinPts* is set to 4.

A dynamic ellipse is designed for the searching neighborhood, whose semi-major adjusts with the position of the sample point, and its expression is

$$\frac{(x - x_p)^2}{E_x^2} + \frac{(y - y_p)^2}{E_y^2} = 1 \tag{3}$$

where $(x, y)$ is the position of the lidar point; $(x_p, y_p)$ is the location of the sample point; and $E_x$ and $E_y$ are the semi-major and semi-minor of the ellipse neighborhood, respectively.

Considering the size of the detected region and the range of the point spacings, $E_y$ is designed as a constant, and $E_x$ is adjusted dynamically by the value of $v$:

$$E_y = \alpha \cdot w, \quad E_x = \beta \cdot \begin{cases} w, & v \le w \\ v, & w < v \le L \\ L, & v > L \end{cases} \tag{4}$$

where $w$ is the mesh width in the grid map; $L$ is a parameter related to the detected object; and $\alpha$ and $\beta$ are the linear coefficient of semi-minor and semi-major, respectively, whose values are positive integers.

From (4), $E_y$ is not an unequivocal constant, and $E_x$ does not obey a strict linear relationship with the longitudinal spacing due to the following:

(a) The existence of a minimum point spacing, because these points were projected onto the mesh of the grid map; thus, $E_y$ is a constant related to $w$.

(b) Similarly, $v$ is saturated by $w$ for the points very close to the lidar, resulting in the existence of a minimum $E_x$.

(c) In the applications, the size of the object to be detected is usually limited, and under-segmentation still easily occurs if a very large neighborhood is applied even at a far distance; hence, an upper bound of $E_x$ is given in (4).

With the aforementioned ellipse and referring to the DSC, the DAC is designed as in Figure 8. Its biggest difference with the traditional DSC is that the fixed circle is replaced with a variable ellipse, increasing the number of parameters. The parameters involved are further analyzed and optimized to better verify the performance of DAC.

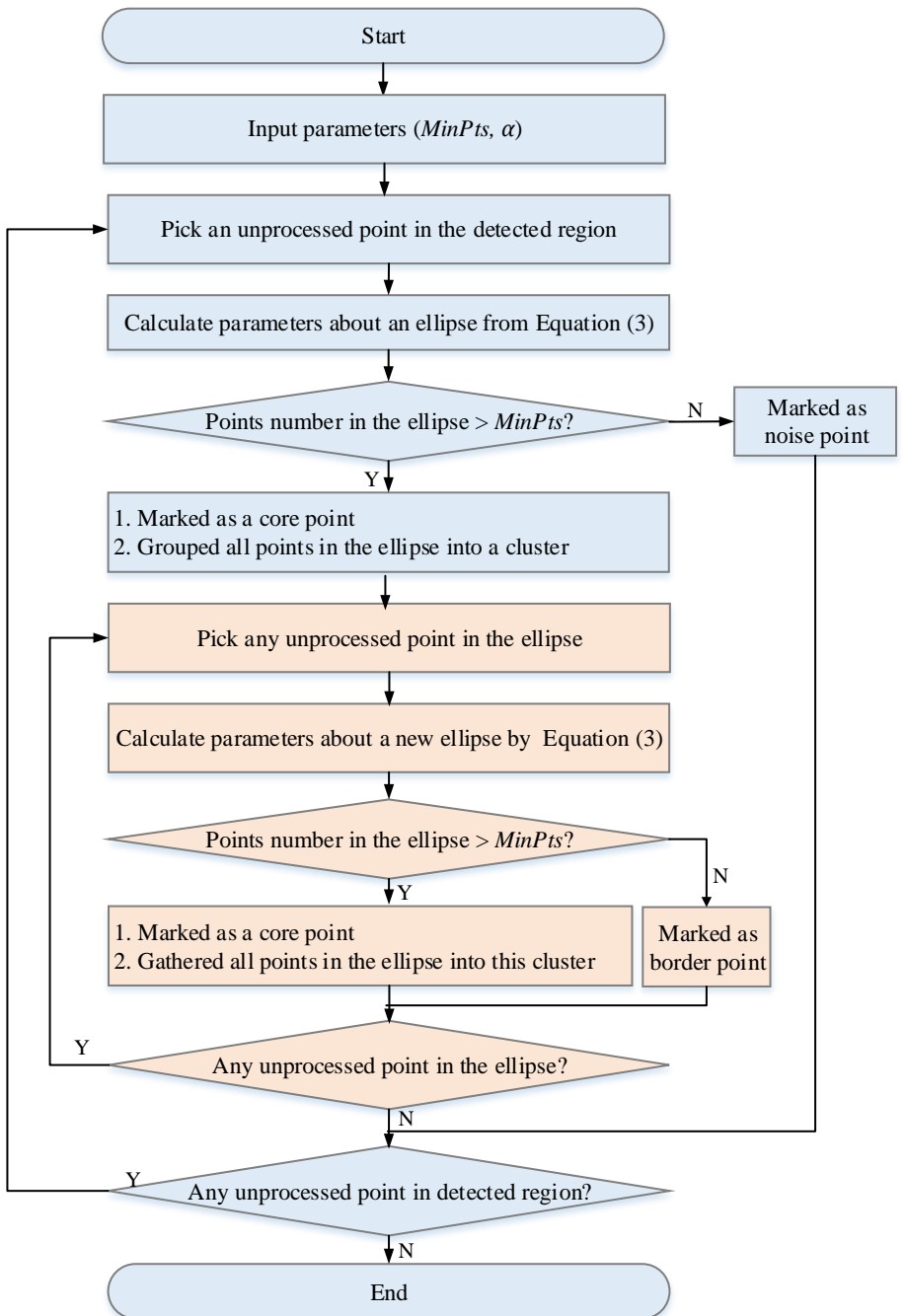

**Figure 8.** Flow chart of DAC method.

### 4. Parameter Design

The parameters of DAC include not only *MinPts* in DSC but also all parameters involved in the axes of the ellipse in (4). Specifically, the inherent attributes of the point cloud, $h$ and $v$, are calculated by the position of the sample point; the mesh width in the grid map, $w$, has been given in the pre-processing stage; the upper boundary of the semi-major, $\beta L$, depends on the maximum length of the detected object according to the design concept from Section 3.2; and the linear coefficients $\alpha$ and $\beta$ need to be studied. Thus, the theoretical relation between *MinPts*, $\alpha$ and $\beta$ is studied, and numerical optimization of parameters is analyzed in the KITTI dataset [34] by considering the comprehensive performance.

### 4.1. Relationship between MinPts and β

The spatial distribution of the point cloud is determined by the properties of the lidar [37]. Thus, the points' number within the neighborhood depends on its area, which is mainly decided by the major axis of the ellipse; hence, β and *MinPts* have a strong positive correlation. To cover as many points belonging to the same object as possible in an ellipse, the points satisfying *MinPts* within this ellipse are evenly arranged on the long axis at the largest point spacing as shown in Figure 9.

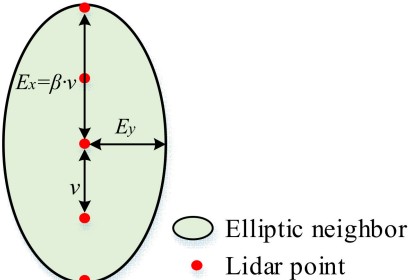

**Figure 9.** Relation between a searching neighborhood and *MinPts*.

From Figure 9, the mathematical relationship between β and *MinPts* is as follows:

$$\beta \geq \lceil (MinPts - 1)/2 \rceil \tag{5}$$

where $\lceil \cdot \rceil$ is the ceiling function. In the applications, the points belonging to the same object may be discontinuous due to the occlusion of other objects, thus increasing β to gather these points into a class correctly. However, for objects close to the lidar, multiple objects are prone to merge if β is too large. Thus, (5) is updated to

$$\beta = \lceil (MinPts + 1)/2 \rceil \tag{6}$$

Compared with the minor axis with a fixed length, the major axis with a variable length has a greater effect on clustering results; thus, the influence of *MinPts* on the results is first analyzed.

### 4.2. Analysis of MinPts

In theory, the smaller the *MinPts*, the easier it is to form effective clusters and further extend the region of the class. However, it also means that when applying a small neighborhood area, the sparse points of the object are easily split into multiple classes. With the increase in *MinPts*, the performance of over-segmentation is improved, but points belonging to multiple objects are mistakenly clustered together in near ranges because of the larger neighborhood, especially for those objects close to each other. The influence of *MinPts* on the clustering results is complex, and the optimal *MinPts* cannot be obtained directly.

An experiment using the KITTI dataset was designed to study the effect of *MinPts* on the overall performance by analyzing the clustering results (such as under-segmentation, over-segmentation, missed detection and correct detection [38]). During the experiment, $E_y$ was set to 0.2 m and initial *MinPts* was 3 [39], and the comparison result is shown in Figure 10.

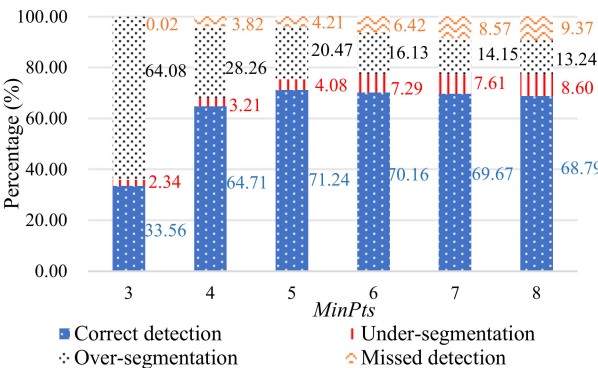

**Figure 10.** Effect of *MinPts* on clustering results.

As shown in Figure 10, the percentage of over-segmentation decreases with the increase in *MinPts*, because the neighborhood area is expanded, and multiple subclasses belonging to the same object are merged correctly. However, this can mistakenly combine multiple objects in the near region, increasing the number of under-segmentation. For an object with few points, such as a pedestrian at a far region, the points' number is still too small to be considered as noise; thus, these objects are missed in detection. With the increase in *MinPts*, the number of these objects increases, increasing the rate of missed detection. From the comprehensive results in Figure 10, the overall performance is the best when *MinPts* is set to 5.

*4.3. Analysis of α*

The mentioned KITTI dataset was still applied for analysis of $\alpha$. During the experiment, *MinPts* = 5 and $\beta$ = 3 from the experimental result in the previous section, and $\alpha$ was set as 1 to 5 because $E_y$ is not always the semi-minor axis of the ellipse, which was defined for convenience. The comparison result is shown in Figure 11.

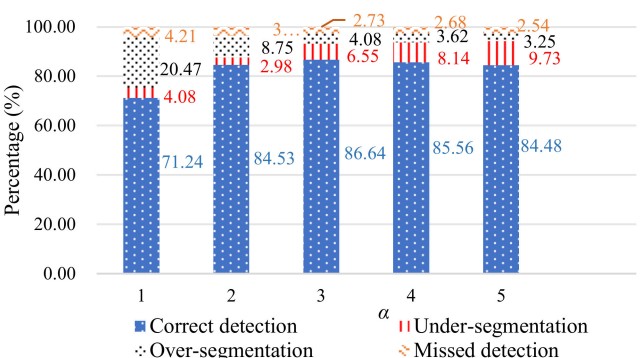

**Figure 11.** Effect of $\alpha$ on clustering result.

The neighborhood area increases with the increase of $\alpha$, which improves the probability that points are effectively clustered. The point spacing is larger in a far range, and if a smaller neighborhood is used for clustering, an object is wrongly segmented into multi-objects or some points are regarded as the noise. In this condition, by increasing the neighborhood area, these points of the same object gradually merge together correctly; thus, the percentages of over-segmentation and missed detection are reduced.

Theoretically, multiple objects are easily merged using a large neighborhood; that is, the number of under-segmentation should increase. However, the curve of under-segmentation is not monotone, and a valley appears at $\alpha$ = 2 (*w* was set to 0.2 m), because the lateral point spacing is less than 0.4 m from Figure 6, except for the extreme values when $\theta$ is close to 90 degrees. In the application, the highest accuracy is achieved at $\alpha$ = 3 by considering the comprehensive performance of the DAC method.

## 5. Experimental Results and Analysis

To verify the improved performance of the DAC, a comparison experiment was carried out on campus, which runs on the onboard PC with an i7-8550U 1.88 GHz CPU. The electric vehicle shown in Figure 12a [40,41] is used as an experimental platform, where IBEO LUX8L-8 is applied to collect the point cloud, and a camera takes pictures to verify the clustering results. The experimental route, shown in Figure 12b, involves dense crowds and complex roads with steep slopes, narrow roads and irregular shapes, which are challenges to correct clustering of the point cloud.

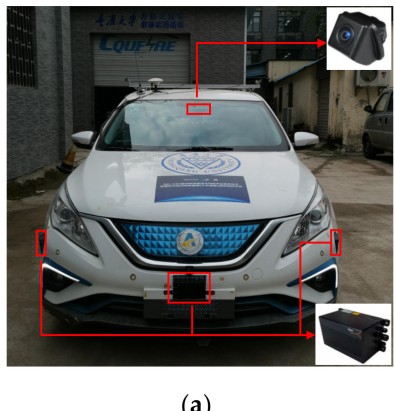　　　　　　　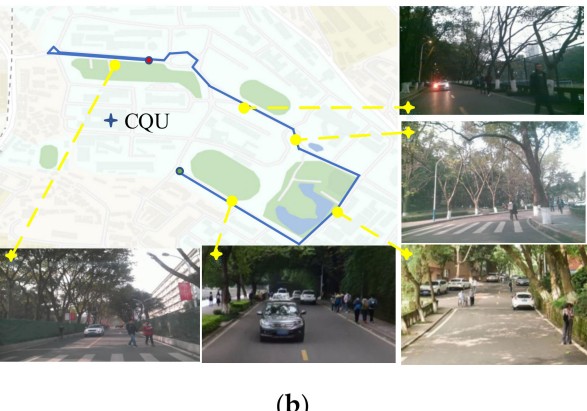

(**a**)　　　　　　　　　　　　　　　　　　　　　　　　(**b**)

**Figure 12.** Experimental platform and scenario. (**a**) Appearance of vehicle and (**b**) diagram of the scenario.

Designing the comparative experiment in this paper, two main elements are considered:

(1) With the on-board unit as the processor, the running time of the methods cannot be too long;

(2) The adaptive ellipse in DAC replaces the fixed circle as the searching neighborhood; thus, the clustering results need to be compared when these two searching neighborhoods are applied.

In the experiment, the partition-based DSC method (SDSC) [27] is used for comparison, because it not only considers the spatial distribution of the point cloud but also still applies the traditional DSC within each sub-region. To ensure the reliability of the experiment, the relevant parameters are first designed. The parameters of DAC are shown in Table 1, following the selection principle of parameters from Section 4.2. When designing the SDSC method, the detected region was divided into three sub-regions based on the density histogram, and each of the parameters is calculated by k-dist [42] as shown in Table 2.

**Table 1.** Parameters of DAC.

| *MinPts* | $\beta$ | $\alpha$ |
|---|---|---|
| 5 | 3 | 2 |

**Table 2.** Parameters of SDSC.

| Partition | Longitudinal Distance (m) | *MinPts* | Neighbor Diameter (m) |
|---|---|---|---|
| 1 | [50, maximum) | 3 | 1.95 |
| 2 | [20, 50) | 5 | 1.06 |
| 3 | (0, 20) | 7 | 0.48 |

Time, weather and other conditions affect the movement of people, vehicles etc.; thus, the experimental vehicle drove along the route several times, and the statistical results of the clustering and running time are shown in Table 3. The accuracy of SDSC with region division is only 74.46%, while that of DAC where the neighborhood varied with the sample

point is 86.54%. Compared with SDSC, the percentages of all wrong clustering in DAC are reduced, with percentages of over-segmentation, under-segmentation and missed detection decreased by 50.58%, 54.85% and 7.43%, respectively. However, DAC runs longer than SDSC, because it takes longer to calculate the distance between two adjacent points. In a practical application, it is feasible to sacrifice the response time to improve the clustering accuracy of DAC, and the main reasons are as follows:

(a) Compared with SDSC, the improvement rates of over-segmentation and under-segmentation in the DAC method are more than half, which provides a good foundation for correctly estimating the size and location of objects [8].

(b) The highest frequency of the IBEO fusion device is 25 Hz, and the DAC does not have a computational delay.

**Table 3.** Statistical results of the two methods.

| Method | Corrected Detection | Over-Segmentation | Under-Segmentation | Missed Detection | Time (ms) |
|---|---|---|---|---|---|
| SDSC | 74.46% | 9.43% | 12.89% | 3.22% | 21 |
| DAC | 86.54% | 4.66% | 5.82% | 2.98% | 38 |

To illustrate the performance advantages of DAC in over-segmentation and under-segmentation more intuitively, a typical scenario is shown in Figure 13, in which two pedestrians walk alongside a stationary vehicle. For the pedestrians who are close to each other, the difference between intra-class distance and inter-class distance is not extremely significant. The circular neighbor with a fixed radius covers more blank areas in the SDSC, causing the pedestrians to be wrongly merged. In the DAC, fewer blank areas are covered using a variable ellipse, in which the major axis is determined by the spacing within a pedestrian, and the minor axis is not affected by its value; thus, these points are correctly split into two classes, and the correct segmentation of pedestrians is achieved accordingly.

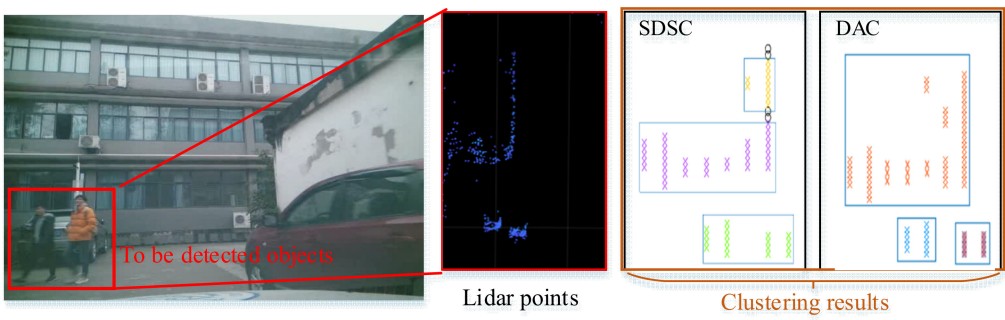

**Figure 13.** Comparison of results in a typical scenario.

For the gray vehicle in Figure 13, the point spacing changes due to the blocking of pedestrians. The vehicle rear is short and is blocked by two pedestrians, with a relatively uniform arrangement of points in the lateral direction appearing, while the vehicle side is longer and is sheltered by one person; thus, the points in this direction change more obviously. In this detected region, SDSC cannot adapt to the fluctuation of point spacing because of the circle with a fixed radius as the neighborhood shape; thus, some points in the longitudinal direction are regarded as noise. Furthermore, the contour points of the vehicle no longer continue due to these noise points, resulting in the vehicle being segmented into two objects. However, in DAC, not only the length of the major axis is adjusted with the sample point, but also the update of the neighborhood area is continuous, tolerating a certain degree of variation; thus, all these points gathered together achieve correct detection.

## 6. Conclusions

In this paper, we propose a new clustering method for object detection using lidar, which effectively improves the clustering performance by considering the spatial distribution of the point cloud. The theoretical, numerical and comparative results show the following:

(a) The point cloud density is significantly different in different ranges and directions, which is fully considered when designing the neighborhood shape.

(b) The designed elliptic neighborhood with adaptive adjustment can improve the clustering performance in all detection regions, while traditional DSC cannot balance the clustering results well between far and near ranges due to the non-uniformity of the point cloud.

(c) In relatively complex environments, DAC has more advantages than SDSC in improving the performance of over-segmentation and under-segmentation.

**Author Contributions:** C.L. wrote the article and assisted with the experiment; B.Z. designed the experiment and analyzed the data; X.H. assisted in the data analysis during the experiment; F.G. provided theoretical guidance and revised the paper. All authors have read and agreed to the published version of the manuscript.

**Funding:** This work was supported in part by the Natural Science Foundation of Chongqing under grant cstc2019jcyj-zdxmX0018 and the Sichuan Science and Technology Program under grant 2020YFSY0070.

**Conflicts of Interest:** The authors declare no conflict of interest.

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
