# Peer review of "A New Density-Based Clustering Method Considering Spatial Distribution of Lidar Point Cloud for Object Detection of Autonomous Driving"

_electronics, doi:10.3390/electronics10162005_

Round 1

Reviewer 1 Report

This is an interesting paper which introduces a new methodology which proposes incorporating additional adaptability in clustering with the aim of improving surrounding object recognition as compared to the traditional density-based spatial clustering (DSC) technique. Overall, the authors have done a fair job at describing the proposed concept, with adequate text and descriptions as needed, together with the experiment design and the improvement through measured data. 

          Coming to the specifics, it is not quite clear (line 117, 118) as to how the authors came to the conclusion that a ‘circular neighborhood causes under-segmentation more easily than an ellipse’, on the basis of a limited dataset (two photographs). Is a circular neighborhood always worse ? Please elaborate this further to improve the reader’s understanding, perhaps with an expanded image / graphic dataset, along with accompanying text.

          The experimental results suggest almost a 2X increase in the computation time Vs. the standard approach. In case of an emergency, fast-response time is often a necessity. Is the proposed technique trading off safety with correctness of detection ? For safe driving, why is it not just sufficient for Lidar to detect the object ? Also, why is a precise detection needed at the expense of response time ? Please elaborate on these further.

          Some typos / sentence structing should be corrected, for instance, lines 10, 11, ‘which greatly degrades the clustering performance of the traditional density-based spatial clustering of application with noise (DSC)’, recommend moving (DSC) to after ‘density-based spatial clustering’. Furthermore, please simplify this line, as it is hard to follow (first line of the abstract). In general, recommend a careful reading to correct these and others.

Author Response

Q1: Coming to the specifics, it is not quite clear (line 117, 118) as to how the authors came to the conclusion that a ‘circular neighborhood causes under-segmentation more easily than an ellipse’, on the basis of a limited dataset (two photographs). Is a circular neighborhood always worse? Please elaborate this further to improve the reader’s understanding, perhaps with an expanded image/graphic dataset, along with accompanying text.

Answer: We are sorry that we did not express clarify. In the revised manuscript, we have re-described the analysis of Figure 4 and marked it by “Revision for Reviewer 1, Q1”.

“In Figure 4 (a), the vehicle is detected correctly when it is 3 m away from the lidar, but it is wrongly classified into three objects when it lies at 14.2 m due to the distance between the points becoming sparse. In Figure 4 (b), a woman stands to close the vehicle and they are classified into the same object wrongly. As can be seen from the enlarged view, their points are very close, and a smaller searching radius is required to distinguish them. Considering that the point spacing between objects is different from that within objects, an anisotropic shape may also distinguish them correctly. To compare the clustering effects of isotropic and anisotropic shapes, circle and ellipse, whose diameter is the same as the long axis, are applied to the two objects in the enlarged image, and the result shows that the circular neighborhood is more prone to un-der-segmentation than the ellipse.”

Q2: The experimental results suggest almost a 2X increase in the computation time Vs. the standard approach. In case of an emergency, fast-response time is often a necessity. Is the proposed technique trading off safety with correctness of detection? For safe driving, why is it not just sufficient for Lidar to detect the object? Also, why is a precise detection needed at the expense of response time? Please elaborate on these further.

Answer: Thank you very much for your comments. The proposed method does improve the correctness of clustering at the expense of response time, because accurate object clustering is the basis for estimating the size and location of objects, as stated in the last sentence of the first paragraph of the Introduction, which is very important for path planning and behavioral decision. Thus, in the new version, the time comparison and practical analysis of DAC are re-written, and marked by “Revision for Reviewer 1, Q2”.

“However, DAC runs longer than SDSC, because it takes longer to calculate the distance between two adjacent points. In practical application, it is feasible to sacrifice the response time to improve the clustering accuracy of DAC, and the main reasons are:

(a) Compared with SDSC, the improvement rates of over-segmentation and un-der-segmentation in the DAC method are more than half, which provides a good foundation for correctly estimating the size and location of objects [7].

(b) The highest frequency of IBEO fusion device is 25 Hz, and the DAC does not have a computational delay.”

       Regarding the question about “For safe driving, why is it not just sufficient for Lidar to detect the object?”, the related answer is as follows:

       First, all descriptions and experimental analysis in this paper are based on lidar; second, the fusion pattern of lidar and camera is widely used in object detection nowadays, mainly for these two reasons:

  • The higher the line beam of lidar, the more accurate environmental information can be obtained. Accordingly, the cost is more and more expensive, which cannot be widely used in commercial vehicles.
  • Due to the unique constraints of point cloud spatial distribution, the object detection technology using lidar needs to be further improved, while the maturity of image detection is much higher than that of lidar, so the fusion mode can obtain better detection performance at present.

Q3: Some typos / sentence structuring should be corrected, for instance, lines 10, 11, ‘which greatly degrades the clustering performance of the traditional density-based spatial clustering of application with noise (DSC)’, recommend moving (DSC) to after ‘density-based spatial clustering’. Furthermore, please simplify this line, as it is hard to follow (first line of the abstract). In general, recommend a careful reading to correct these and others.

Answer: Thank you very much for your suggestions. Firstly, density-based spatial clustering of application with noise is a fixed collocation, generally referred to as DBSCAN, which is abbreviated as DSC in this paper. In addition, another reason why this paragraph is difficult to understand maybe that punctuation is written incorrectly, which has been changed and marked by “Revision for Reviewer 1, Q3” in the revised manuscript. 

Reviewer 2 Report

This paper proposes to use LIDAR to detect nearby object in transportation environment. The motivation is good and reasonable and design is appropriate, and literature is extensive.

However the clarify needs significant improvements, for example what is DSC in the beginning (people can guess what DAC means), and KITTI etc, not assuming a reader is familiar with all underlying existing methods and dataset.

Also, the paper should clearly highlight the real novelty, is it method or data or an implementation of prototype? What do you mean 'according to theory difference'? Actually the calculation of object crash follows some classical and some relative new mathematical models (see a recent paper 10.1109/TITS.2021.3093714 who did a great job reviewing literature though the paper was about pedestrian conflict not cars), this is pretty sensitive to LIDAR environment with assistance of cameras. 

Regarding experiment results, it seems the comparison with SOTA is not strongly supportive to the significance of the work. There is a question which work would become a benchmark. 

Author Response

Q1: The clarify needs significant improvements, for example what is DSC in the beginning (people can guess what DAC means), and KITTI etc, not assuming a reader is familiar with all underlying existing methods and dataset.

Answer: Thank you very much for your advice. For the abbreviation DSC, its full name was written when it first appears in both the abstract and the text, i.e. lines 11 and 53. For the KITTI dataset, we added its description and marked it by “Revision for Reviewer 2, Q1” in the revised version.

Q2: Also, the paper should clearly highlight the real novelty, is it method or data or an implementation of prototype? What do you mean 'according to theory difference'? Actually the calculation of object crash follows some classical and some relative new mathematical models (see a recent paper 10.1109/TITS.2021.3093714 who did a great job reviewing literature though the paper was about pedestrian conflict not cars), this is pretty sensitive to LIDAR environment with assistance of cameras.

Answer: Thank you for pointing this out. This paper highlight the method novelty, and related sentence is rewritten and marked by “Revision for Reviewer 2, Q2-1”.

“For the object detection of lidar in the autonomous driving systems, a DSC-based adaptive clustering (DAC) method is proposed in this paper by considering the spatial distribution of point cloud, to improve the overall clustering results of traditional DSC.”

       For the sentence “according to theory difference”, the previous description was not clear, which has been corrected and marked by “Revision for Reviewer 2, Q2-2”

       “According to different clustering principles, the traditional clustering methods are mainly divided into partition-based method [12-14], hierarchical method [15][16] and density-based method [17-19].”

       For the last question, we did not mention it, so we add the relevant content in the revised version and marked it by “Revision for Reviewer 2, Q2-3”.

“It’s obvious that automated vehicles have a profound impact on human social activities and lifestyles [1][2], in which object detection plays an important role to ensure driving safety, especially the location estimation and trajectory prediction of objects [3].”

Q3: Regarding experiment results, it seems the comparison with SOTA is not strongly supportive to the significance of the work. There is a question which work would become a benchmark.

Answer: Thank you very much for your comments. In the problem description section, two possible reasons for the poor performance of traditional DSC have been discussed, (1) The neighborhood shape is constantly adjusted with the sample point, and (2) The distribution of point spacing is different in two directions. The former problem has been found and improved by many people (for example SDSC), while the latter one has attracted little attention. In this paper, two main elements are considered to design the comparison experiment as follows:

       (1) The on-board unit is applied as the processor, thus the computation time for these methods cannot be too long to ensure that there is no computational latency.

       (2) The adaptive ellipse in the proposed method (DAC) replaces the fixed circle in traditional DSC as the neighborhood shape, so their clustering results are compared to illustrate the improvement effect of DAC.

       Thus, in the comparison experiment, DAC is compared with the improved scheme SDSC, which has a fixed radius in each sub-region and considering the spatial distribution of point cloud.

       In the experimental analysis, the running time of both methods is lower than the IBEO fusion time, which satisfies the application requirements, i.e., the first design condition. The percentage of the different clustering results for the two methods is compared to illustrate the accuracy of the algorithms, that is, the second condition. The wrong results of DAC are all improved in varying degrees compared to SDSC, and the advantages of DAC are further described and analyzed with a typical scene.

Based on the above analysis, we added the experimental design purpose in the revised manuscript and marked it by “Revision for Reviewer 2, Q3”

“Designing the comparative experiment in this paper, two main elements are considered:

(1) With the on-board unit as the processor, the running time of the methods cannot be too long;

(2) The adaptive ellipse in DAC replaces the fixed circle as the searching neighborhood, so the clustering results need to be compared when these two searching neighborhood are applied.

In the experiment, the partition based DSC method (SDSC) [26] is used for comparison, because it not only considers the spatial distribution of the point cloud, but also still applies the traditional DSC within each sub-region.”